# Type 2 Diabetes Risk Perception and Health Behaviors Among Women with History of Gestational Diabetes Mellitus: A Retrospective Analysis

**DOI:** 10.3390/nu17213360

**Published:** 2025-10-25

**Authors:** Allyson Malone, Tristan D. Tibbe, Norman Turk, Obidiugwu Kenrik Duru, Lauren E. Wisk, Carol Mangione, Jessica Page, Samuel C. Thomas, Amanda Vu, Ruth Madievsky, Janet Chon, Felicia Cheng, Sandra Liu, Richard Maranon, Jacob Krong, Ashley Krueger, Christina S. Han, Keith Norris, Tannaz Moin

**Affiliations:** 1Division of General Internal Medicine-Health Services Research, Department of Medicine, University of California, Los Angeles, CA 90024, USA; 2Department of Medicine Statistics Core, UCLA Health, Los Angeles, CA 90024, USA; 3Department of Maternal-Fetal Medicine, Intermountain Health, University of Utah Health, Salt Lake City, UT 84143, USA; 4Department of Internal Medicine, Intermountain Health, Murray, UT 84017, USA; 5Healthcare Delivery Institute, Office of Research, Intermountain Health, Murray, UT 84107, USA; 6Division of Maternal-Fetal Medicine, Department of Obstetrics and Gynecology, University of California, Los Angeles, CA 90095, USA; 7Division of Endocrinology Diseases, Diabetes and Metabolism, Department of Medicine, University of California, Los Angeles, CA 90024, USA; 8HSR and CSP NODES, VA Greater Los Angeles Healthcare System, Los Angeles, CA 90073, USA

**Keywords:** women’s health, diabetes prevention, gestational diabetes, nutrition, physical activity

## Abstract

**Background/Objectives**: History of gestational diabetes mellitus (GDM) is a strong risk factor in the development of type 2 diabetes (T2D). We sought to examine the association between perceived risk of developing T2D and relevant health behaviors in this population. **Methods**: We analyzed self-reported survey items and objective electronic health record data of participants in the Gestational diabetes Risk Attenuation for New Diabetes (GRAND) Study, a multisite randomized control trial testing the effectiveness of shared decision-making for women with elevated body mass index (BMI), prediabetes and history of GDM. Data on demographics, health behaviors, and perceived T2D risk were self-reported. We ran four regression models to study the association between women’s perceived risk of developing T2D and four key health behaviors: (1) physical activity, (2) consumption of sugar-sweetened beverages, (3) consumption of ultra-processed foods, and (4) consumption of meals prepared outside the home. All models were adjusted for age, race, ethnicity, income, HbA1c, BMI, family history of T2D, and study arm. **Results**: Our sample included 242 women who on average were 41 years old (±6 years) with BMI of 32.7 (±6.9 kg/m^2^). Perceived risk of developing T2D was not significantly associated with physical activity, consumption of sugar-sweetened beverages, ultra-processed food consumption, or meals prepared outside of the home. Higher BMI was significantly associated with increased consumption of sugar-sweetened beverages (OR 1.05, 95% CI 1.01–1.10), but not other health behaviors. **Conclusions**: We found perceived risk of developing T2D was not independently associated with four key health behaviors. Women with GDM are at high risk of developing T2D and may benefit from tailored or more intensive strategies promoting health behavior changes shown to lower T2D risk.

## 1. Introduction

Women with a history of gestational diabetes mellitus (GDM) are at high risk of developing type 2 diabetes mellitus (T2D) as compared to the general population [1]. Interventions to prevent the development of T2D among women with history of GDM are well-studied. Randomized clinical trials have shown that metformin or intensive lifestyle intervention, such as the Diabetes Prevention Program (DPP), are both effective in lowering T2D risk in this population [2]. The DPP emphasizes a variety of lifestyle changes to reduce incident T2D risk, including increasing physical activity and dietary modifications.

The DPP emphasizes ≥150 min of moderate physical activity per week and studies support many types of exercise as beneficial in reducing T2D risk [3]. While the DPP dietary goals focus on reducing overall caloric and fat intake, numerous additional studies have advanced our understanding of beneficial dietary modifications in diabetes prevention [4]. The importance of reducing intake of ultra-processed foods (UPF), fast food or meals prepared outside the home, and sugar-sweetened beverages (SSB) has become more evident over the last two decades [5,6,7]. UPF consumption is associated with overweight/obesity and other cardiovascular complications [8]. Frequent consumption of meals prepared at home is associated with lower risk of T2D, while fast-food consumption is associated with weight gain and insulin resistance [9,10]. Regular SSB consumption is associated with greater insulin resistance and higher risk of developing T2D [11]. Notably, SSB consumption increases the risk of developing T2D in a dose-dependent manner, suggesting a potential causal relationship [12].

While GDM is one of the strongest risk factors for T2D, many women with GDM are unaware of their risk [13,14]. Even when women are aware that GDM is a population-level risk factor for T2D, many who have GDM do not perceive themselves to be at high risk of developing T2D [15,16,17]. Our group previously found that nearly 1/3 of women with a history of GDM were unaware that GDM was a risk factor for T2D and nearly half thought themselves to be at low risk [13]. Risk perception is important since both the Health Belief Model and behavior motivation hypothesis posit that risk perception of developing a disease predicts participation in health behaviors intended to prevent that disease [18]. In one study, patients with T2D that had high risk perception of their disease were more likely to engage in diabetes self-care behaviors, supporting the hypotheses above [19]. In a more recent study, women with history of GDM who had higher perceived risk reported they were more likely to plan to change their behavior, though actual behavioral change was not studied [20]. As these frameworks have been used to study other disease states, we were interested in further exploring the relationship between risk perception and health behaviors in our population. Therefore, our study aimed to examine the association between perceived T2D risk and four key health behaviors: (1) physical activity, (2) SSB consumption, (3) UPF consumption, and (4) consumption of meals prepared outside of the home. We hypothesized that women reporting higher personal risk perception of developing T2D would engage in healthier behaviors, namely being more physically active and consuming less SSBs, UPFs, and meals outside of the home.

## 2. Materials and Methods

We analyzed 2021–2024 data collected in the Gestational diabetes Risk Attenuation for New Diabetes (GRAND) study. GRAND is an ongoing multicenter randomized controlled trial (RCT) testing the effect of a T2D prevention shared decision-making intervention on weight loss and uptake of T2DM prevention interventions in women with overweight/obesity, prediabetes and a history of GDM receiving care at either University of California, Los Angeles (UCLA) or Intermountain Health (IH). Details of the GRAND study protocol can be found elsewhere [21]. The GRAND study was approved by the UCLA IRB (IRB#20-001558).

GRAND study eligibility criteria included women between 18 and 54 years of age with a history of GDM (based on ICD-10 diagnoses or diagnostic 1 or 2-step oral glucose tolerance testing [OGTT]) and prediabetes within the last 36 months (based on labs including HbA1c 5.7–6.4%, fasting glucose 100–125 mg/dL, or 2-h glucose 140–199 mg/dL); BMI ≥ 25 kg/m^2^ [or >23 if Asian American]). Additionally, participants had to have a health care encounter (any provider visit or any labs) within the last 12 months at UCLA or IH. Women with any prior diagnosis of T2D (HbA1c > 6.5%, ≥2 fasting glucose > 125 mg/dL, or use of anti-glycemic medication within 12 months), chronic kidney disease (eGFR < 45 mL/min), and those planning to get pregnant in the next 12 months, or currently pregnant, were excluded.

An initial EHR data query identified potentially eligible participants who received an invitation letter via mail after PCP notification. Research team members conducted telephone recruitment calls to confirm eligibility and recruit women for enrollment in the study. After informed consent was obtained, participants were randomized to either a shared decision-making (SDM) intervention or usual care. GRAND study participants completed surveys at baseline, 6 months, 12 months, and 24 months.

We collected baseline patient characteristics including age, HbA1c, and BMI from the EHR. GRAND participants completed surveys assessing demographic information, risk perception, and health behaviors. Patients self-reported race, ethnicity, income, and family history of T2D. Personal risk perception of developing T2D was assessed using a single item, while health behaviors were assessed using various survey items. We included 4 survey items that focused on key health behaviors; (1) physical activity (binary yes/no), (2) sugar-sweetened beverage consumption (categorized as none, <1/day, and 1+/day), (3) ultra-processed food consumption (rarely/never, sometimes, and often), and (4) meals prepared outside of the home (see Appendix B). Our sample consisted of patients who had responses to all of the covariates of interest. In order to fill in missing race and ethnicity data at the baseline survey, follow-up surveys also asked patients about race and ethnicity, and their responses were used to input missing baseline values.

We analyzed survey responses regarding health behavior and risk perception from GRAND participants collected at 6 months. All participants were aware of GDM as a strong risk factor for T2D development as they were either given supplemental educational material (control group) or discussed T2D prevention with a trained clinician (intervention group). Our primary outcomes of interest were the four health behaviors described above. We conducted four separate regression models to examine the association between perceived risk of developing T2D and each of the four health behaviors of interest. We used a multivariate logistic regression model to examine the physical activity outcome (Model 1), multivariate ordinal logistic regression models to separately examine SSB and UPF consumption (Models 2 and 3), and a multivariate linear regression model to examine consumption of meals prepared outside the home (Model 4). All models were adjusted for age, race, ethnicity, income, HbA1c, BMI, family history of T2D, and study arm. Analyses were performed between 2022 and 2024 using SAS version 9.4.

## 3. Results

Our analytic cohort included 242 women enrolled in the GRAND trial who completed baseline and six-month follow-up surveys between 2021 and 2024. On average, women were 41 years old (range 22–54), with mean HbA1c 5.9% (±0.2), and mean BMI 32.7 kg/m^2^ (±6.9 kg/m^2^). Half (50%) reported a family history of diabetes. Participants self-identified as 28% Asian American and Pacific Islander (AAPI), 5% Black, 22% Latino, 3% Multiracial/other, and 42% White. Our participants resided in rural, suburban, and urban settings and had received health care from one of two large hospital systems in the Western United States.

On self-reported survey items assessing perceived chance of developing diabetes, 32% stated they had a slight chance of developing T2D, 45% reported a moderate chance, and 24% endorsed having a high chance (Table 1). Most participants (82%) reported being physically active. Over one-third (36%) reported consuming no sugar-sweetened beverages on a weekly basis, nearly half (48%) consumed less than one sugar-sweetened beverage per day, and 14% consumed one or more sugar-sweetened beverages daily. When asked about frequency of consuming ultra-processed foods, 34% reported rarely/never, 49% reported sometimes, and 17% reported often. Participants consumed a median of 3 meals prepared outside of the home in the last seven days (IQR 2–5) (Table 1). We included participant characteristic data stratified by perceived risk in the Appendix A. 

In our multivariate models, we did not find a significant association between personal risk perception of developing T2D and physical activity (slight vs. high risk perception OR 1.01, 95% CI = 0.37–2.81, moderate vs. high risk perception OR 0.98, 95% CI = 0.39–2.47), consumption of sugar-sweetened beverages (slight vs. high risk perception OR 1.09, 95% CI = 0.55–2.19, moderate vs. high risk perception OR 0.96, 95% CI = 0.50–1.85), consumption of ultra-processed foods (slight vs. high risk perception OR 0.54, 95% CI = 0.26–1.10, moderate vs. high risk perception OR 0.74, 95% CI = 0.38–1.43), or consumption of meals prepared outside the home (slight vs. high risk mean difference −0.34, 95% CI = −1.55–0.86, moderate vs. high risk mean difference 0.28, 95% CI = −0.85, 1.40) (Table 2). However, participants with higher BMI had higher odds of consuming sugar-sweetened beverages (OR 1.05, 95% CI = 1.01, 1.10) than participants with lower BMI. Women who identified as AAPI or Latino as compared to identifying as White were more likely to consume sugar-sweetened beverages (OR 2.62, 95% CI = 1.29–5.33 and OR 2.59, 95% CI = 1.32–5.05, respectively). When compared to participants in the highest income bracket (>$200,000), participants who reported an income of less than $100,000 on average consumed significantly less meals prepared outside the home (adjusted mean difference −1.50, 95% CI = −2.71, −0.29) and had significantly lower odds of being inactive (OR 0.29, 95% CI = 0.10–0.86).

## 4. Discussion

Our study found no significant associations between women’s perceived risk of developing T2D and four key health behaviors shown to impact T2D risk (physical activity, consumption of sugar-sweetened beverages, consumption of ultra-processed foods, and consumption of meals prepared outside the home). Furthermore, nearly one-third of our participants believed they were at low risk of developing T2D. Certain health theories, such as the health belief model and the behavior motivation hypothesis, have posited that perceived risk of developing an illness may influence engagement in preventative health behaviors [18,22]. In fact, one study found that in adults with prediabetes, awareness of a personal prediabetes diagnosis was associated with increased likelihood in participating in lifestyle interventions to prevent the onset of T2D [23]. However, our study indicates that the relationship between risk perception and health behaviors may be more nuanced. Though we hypothesized that women with higher perceived risk would partake in healthier behaviors based on the Health Belief Model, our findings support the theory that perceived risk and behavior change is a bidirectional relationship [24]. It is possible that our study participants may have evaluated their own health behaviors and incorporated their assessment when considering their personal risk of developing type 2 diabetes. Our findings are particularly concerning since our study focused on women with overweight or obesity, prediabetes and history of gestational diabetes, who are at very high risk of developing T2D. We did find that higher BMI was significantly associated with increased consumption of sugar-sweetened beverages. SSB reduction interventions at the individual level have been found to be relatively easy to implement, and reducing intake of SSBs by even one drink per day has contributed to reduced risk of developing type 2 diabetes [25].

It is also important to note that our understanding of modifiable T2D risk factors has continued to evolve. While physical activity and weight loss have been studied robustly as modifiable risk factors to prevent T2D, the study of other health behaviors are newer focuses of prevention. Consumption of ultra-processed foods, for example, has been recently linked to increased T2D risk [3,5,7,26]. Ultra-processed food consumption continues to rise in the United States. One study found that on average, more than half of American adults’ daily caloric intake came from ultra-processed foods [27]. Studies have shown that pregnant and post-partum women consume more ultra-processed foods when compared to non-pregnant women, which may further contribute to the risk of developing T2D amongst an already at-risk population [28]. Thus, as our understanding of all modifiable T2D risk factors continues to grow, we need interventions that address a wider array of behaviors including the importance of reducing ultra-processed food and sugar-sweetened beverage consumption.

Over the last 30 years, numerous studies have demonstrated that GDM is associated with significant T2D risk, with one systematic review finding that women with history of GDM have a seven-fold increased risk of developing T2D [1,29,30]. Our participants were aware that GDM is a strong risk factor for development of T2D as they either received educational materials or an SDM intervention that both included counseling about GDM as a risk factor. Even with this knowledge, a majority of our participants (76.4%) thought themselves to be at only slight or moderate risk of developing T2D. Additionally, our results suggest that even when women perceive themselves to be at high risk of developing T2D, their high risk perception is not associated with engagement in healthy behaviors that prevent progression to T2D. These findings are in contrast to the broader population of people with prediabetes, who have been shown be more likely participate in weight reduction and exercise behaviors when they are aware of their prediabetes diagnosis [23].

Though awareness can be increased by education interventions, our study suggests education alone on risk of T2D may not be adequate to increase participation in prevention behaviors amongst our study population. Some studies have shown that even when women are aware of their increased risk, they may face many barriers when trying to engage in preventative lifestyle interventions. For example, women with a history of GDM who were two years post-partum have cited fatigue, maternal attachment, and increased household responsibilities as factors that limit their engagement in preventative health behaviors [31]. So, in addition to simply raising awareness about their T2D risk, we also need to tailor lifestyle interventions to better meet women’s needs.

Our study also has some limitations. Though our participants were from diverse racial/ethnic groups and recruited from two regional health systems, our results may not be generalizable to all women with history of GDM. Our behavioral outcomes at 6 months follow-up were self-reported, which could have resulted in some reporting bias. However, we ensured that questions about health behaviors were standardized and delivered by trained study staff to minimize bias (Appendix B). Additionally, though we controlled for control arm vs. intervention arm in our analysis, the impact of our intervention was not the focus of this particular project. While we believe that an SDM intervention among this population could affect both risk perception of developing T2D and various health behaviors, we will be exploring the impact of the SDM intervention in other research works. Our work includes a sample size of 242 women enrolled into the GRAND study, which limits our ability to detect associations between covariates, perceived risk, and health behaviors. We also acknowledge our participants reside primarily in the Western United States which limits generalizability to the general US population and the population of women with history of GDM around the world.

## 5. Conclusions

In summary, we found no association between four key health behaviors and perceived risk of developing T2D amongst women at high risk of developing T2D, but increased BMI was associated with higher consumption of sugar-sweetened beverages. The relationship between perceived risk and preventative health behaviors is complex and necessitates further investigation. Lack of awareness of T2D risk was noted among 1/3 of our study participants, suggesting the need for interventions that can increase understanding of T2D risk. Future studies should examine how interventions can be personalized and incorporate evolving understanding of both the mediators and barriers to key health behaviors in women with a history of GDM and their relationship to T2D.

## Figures and Tables

**Table 1 nutrients-17-03360-t001:** Characteristics of study cohort.

Participant Characteristics	*n* = 242
Age (SD)	40.8 years (6.0)
HbA1c (SD)	5.9% (0.2)
BMI (SD)	32.7 kg/m^2^ (6.9)
Trial arm	
Control group	120 (49.6%)
SDM group	122 (50.4%)
Race and ethnicity	
AAPI	68 (28.1%)
Black	11 (4.6%)
Hispanic/Latino	54 (22.3%)
Multiracial/Other	8 (3.3%)
White	101 (41.7%)
Family history of diabetes	
No	120 (49.6%)
Yes	122 (50.4%)
Annual household income	
<$100,000	80 (33.1%)
$100,000–$199,999	79 (32.6%)
≥$200,000	69 (28.5%)
Don’t Know/Decline to answer	14 (5.8%)
Personal risk perception of T2D	*n* = 242
Slight chance	77 (31.8%)
Moderate chance	108 (44.6%)
High chance	57 (23.6%)
Physical activity	*n* = 241
No	42 (17.4%)
Yes	199 (82.2%)
Sugar-sweetened beverage consumption	*n* = 238
None	86 (35.5%)
<1/day	117 (48.4%)
≥1/day	35 (14.5%)
Ultra-processed food consumption	*n* = 241
Rarely/never	82 (33.9%)
Sometimes	118 (48.8%)
Often	41 (16.9%)
Consumption of meals prepared outside the home in last 7 days	*n* = 242
Median (IQR)	3 meals (2–5)

HbA1c = Hemoglobin A1c, BMI = Body Mass Index, SDM group = the participants randomized to the shared decision-making intervention, and AAPI = participants that identify as Asian American or Pacific Islander.

**Table 2 nutrients-17-03360-t002:** Adjusted results: Association of four key health behaviors with perceived risk and covariates.

	Physical Inactivity	Consumption of Sugar-Sweetened Beverages	Consumption of Ultra-Processed Foods	Consumption of Meals Prepared Outside the Home
Characteristic	Adjusted OR (95% CI)	Adjusted OR (95% CI)	Adjusted OR (95% CI)	Adjusted Mean Difference (95% CI)
**Perceived Risk**				
Slight	1.01 (0.37, 2.81)	1.09 (0.55, 2.19)	0.54 (0.26, 1.10)	−0.34 (−1.55, 0.86)
Moderate	0.98 (0.39, 2.47)	0.96 (0.50, 1.85)	0.74 (0.38, 1.43)	0.28 (−0.85, 1.40)
High	—	—	—	—
**Trial Arm**				
Control Arm	**2.12** **(1.03, 4.36)**	1.22 (0.74, 2.01)	1.48 (0.90, 2.43)	**1.13** **(0.27, 1.99)**
SDM Arm	—	—	—	—
**Race and Ethnicity**				
AAPI	1.45 (0.55, 3.78)		0.77 (0.38, 1.54)	0.94 (−0.28, 2.16)
Black	0.55 (0.06, 5.01)	3.26 (0.98, 10.87)	0.67 (0.20, 2.22)	0.84 (−1.29, 2.97)
Hispanic/Latino	1.17 (0.45, 3.05)	**2.59** **(1.32, 5.05)**	0.67 (0.35, 1.29)	0.29 (−0.85, 1.43)
White	—	—	—	—
Other/Multiracial	0.52 (0.06, 4.72)	0.55 (0.12, 2.54)	0.95 (0.20, 4.46)	0.34 (−2.14, 2.81)
**Income**				
<$100,000	**0.29** **(0.10, 0.86)**	0.96 (0.48, 1.95)	0.68 (0.34, 1.38)	**−1.50** **(−2.71, −0.29)**
$100,000–$199,999	0.91 (0.39, 2.09)	1.01 (0.53, 1.91)	0.98 (0.51, 1.86)	−0.71 (−1.83, 0.40)
≥$200,000	—	—	—	—
Don’t Know or Decline to answer	0.97 (0.22, 4.35)	1.06 (0.30, 3.78)	2.52 (0.83, 7.71)	−1.53 (−3.53, 0.46)
**Family History of T2D**				
Yes	0.93 (0.45, 1.93)	0.86 (0.51, 1.43)	1.28 (0.78, 2.12)	−0.85 (−1.74, 0.03)
No	—	—	—	—
**Age**	0.98 (0.92, 1.05)	1.00 (0.95, 1.04)	1.00 (0.96, 1.05)	−0.04 (−0.12, 0.03)
**HbA1c**	3.05 (0.41, 22.89)	0.60 (0.16, 2.19)	2.65 (0.70, 10.07)	0.62 (−1.73, 2.96)
**BMI**	1.04 (0.98, 1.10)	**1.05** **(1.01, 1.10)**	1.04 (0.998, 1.08)	0.01 (−0.06, 0.08)

HbA1c = Hemoglobin A1c, BMI = Body Mass Index, SDM arm = the participants randomized to the shared decision-making intervention, and AAPI = participants that identify as Asian American or Pacific Islander. Bolding indicates statistically significant results (*p* < 0.05).

## Data Availability

The datasets presented in this article are not readily available because data are part of an ongoing study. Requests to access the datasets should be directed to Dr. Tannaz Moin.

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
