# Peer review of "Type 2 Diabetes Risk Perception and Health Behaviors Among Women with History of Gestational Diabetes Mellitus: A Retrospective Analysis"

_nutrients, 2025, doi:10.3390/nu17213360_

Round 1
Reviewer 1 Report
Comments and Suggestions for Authors
This study indicated that preceied risk of developing T2D in GDM cases does not translate into health behaviors. I believe similar results would be obtained not only for GDM but also for the general population. The content lacks novelty. I think it is necessary to incorporate into the survey items whether individuals undergo regular checkups after a GDM diagnosis, as well as changes in glucose tolerance and body weight.
Reviewer 2 Report
Comments and Suggestions for Authors
The work presented in this manuscript investigated perception of T2D risk and health behavior of women who had experienced GDM and who have predic symptoms and(? or?) BMI 25<. The study revealed that the subjects had relatively low risk perception and perceived risk was not associated with health behavior for preventing development to T2D. This study result may be of some help for developing public health strategy for the prevention of T2D for GDM women.
One thing that is not clear to me is that why the authors investigated health behavior of GDM-experienced women though the previous studies have already shown that awareness of GDM experienced women was low. The authors cited several papers on this topic in Introduction section. If you think it normally, then health behavior of those women would not be expected to be good, and, as expected, the present study confirmed it.
Do the authors believe that GDM-experienced women would not be aware the risk but they would paradoxically well behave? If so, please explain the belief with supportive citations. Otherwise, this work seems to be just a confirmation of the things fully predictable.
Other drawbacks of this work include relatively small number of subjects and potential selection bias. Probably, the subjects of this study had better opportunity of obtaining health information or better educated than those who were not eligible in the Project. It would be informative to refer to the characteristics of the subjects in relation to general women in the US because non-US readers are not familiar.
Reviewer 3 Report
Comments and Suggestions for Authors
I commend the authors for addressing a highly relevant and timely topic concerning type 2 diabetes (T2D) prevention in a high-risk postpartum population. The focus on risk perception and its relationship with health behaviors is of great interest and aligns well with current public health priorities. That said, I would like to offer several suggestions that may help improve the manuscript’s conceptual clarity, methodological rigor, and interpretative depth.
-
The study sample comprises exclusively women already at high clinical risk for T2D (e.g., prior GDM, overweight/obesity, and in some cases, prediabetes), which reduces variability in objective risk and limits the ability to detect associations based on subjective risk perception. Furthermore, the study appears to rely on a conceptual framework where perceived risk precedes and potentially drives behavior. However, it is plausible that risk perception is actually shaped by current behaviors. For instance, individuals who recognize they are not engaging in healthy behaviors may perceive themselves as being at higher risk. I encourage the authors to discuss the possibility of reverse causality more explicitly in the manuscript.
-
I also suggest that the authors consider an alternative interpretation of their findings: women who perceive themselves at high risk may do so not primarily because of their clinical history (GDM), but because they are aware of unhealthy behaviors such as poor diet or low physical activity. Conversely, those who report low risk perception may believe they are engaging in healthy behaviors. If self-assessed diet quality or perceived lifestyle health were available, they could provide further insight. This possibility could help explain the lack of observed associations between perceived risk and objectively measured health behaviors.
-
I suggest stratifying Table 1 by perceived risk categories (e.g., low, moderate, high). This would provide readers with a clearer understanding of how sociodemographic and behavioral factors vary across risk perception levels, which is particularly relevant given that risk perception is the main exposure in the study. Such descriptive stratification could also contextualize the adjusted models and clarify potential baseline differences.
-
Given the modest effect sizes expected in behavior-related outcomes, the current sample size may not provide sufficient power to detect associations. This limitation is acknowledged briefly, but could be emphasized further, especially in light of null findings.
-
In the introduction, the authors cite older references (ref 18) to support the assumption that perceived risk drives behavior change (Health Belief Model). I encourage the inclusion of more recent literature that examines this relationship in the context of chronic disease prevention. This would strengthen the theoretical justification for the study.
Overall, this is a well-intentioned and relevant study. Addressing the points above may enhance the clarity and impact of the findings, while also providing a more nuanced understanding of the complex interplay between risk perception and health behaviors in postpartum women.
Round 2
Reviewer 1 Report
Comments and Suggestions for Authors
I fully understand your objectives.
Author Response
Thank you.
Reviewer 2 Report
Comments and Suggestions for Authors
The authors replied in the manner somewhat out of focus of my comments.
For the 1st comment, the point was that why the authors carried out this study though it was known in the previous publications that GDM experienced women are aware the risk of T2D but they do not behave suitably. I commented the results of this study was largely similar to the findings of the previosu studies. In other words, what is the novel, original element(s) of this work?
In the 2nd comment, what was asked was to explain briefly the characterisics of the study population for non-US readers. I did not criticize the generalizabitlity of this work: no information on the charactristics of the study population hampered comments on the generalizability.
Author Response
The authors replied in the manner somewhat out of focus of my comments.
For the 1st comment, the point was that why the authors carried out this study though it was known in the previous publications that GDM experienced women are aware the risk of T2D but they do not behave suitably. I commented the results of this study was largely similar to the findings of the previous studies. In other words, what is the novel, original element(s) of this work?
response:Thank you for further clarifying your comments for us. To our knowledge, this is the first study to examine the relationship between risk perception and specific lifestyle behaviors in women with history of GDM. As described in the manuscript, the women in this study were exposed to education about their risk of developing T2D. We found that even women that perceived themselves to be at high risk of developing T2D did not partake in healthier behaviors, suggesting a complex relationship between risk perception and behavior change. Though risk perception and health behavior change has been studied amongst other populations with chronic diseases including patients with T2D, this relationship has not been studied amongst women with history of GDM, and our findings are concerning because these women are the highest-risk group of developing T2D.
While there are several studies that have focused on diabetes risk awareness, including a paper led by our group, no study has focused on specific health behaviors. Our focus on specific behaviors is unique as ultra-processed food and sugar-sweetened beverage consumption are more recent areas of study and focus for prevention, while physical activity has been studied more robustly. Our study emphasizes the importance of intensive and tailored interventions to prevent T2D amongst GDM-experienced women as education alone is likely not sufficient for creating lasting behavioral change.
In the 2nd comment, what was asked was to explain briefly the characterisics of the study population for non-US readers. I did not criticize the generalizabitlity of this work: no information on the charactristics of the study population hampered comments on the generalizability.
response:We have added an additional description on page 4 lines 146-148 “Our participants resided in rural, suburban, and urban settings and had received healthcare from one of two large hospital systems in the Western United States”. We have also included a table of baseline characteristics in the manuscript (Table 1) If there are specific characteristics the reviewer thinks would be helpful, we welcome additional thoughts.
Round 3
Reviewer 2 Report
Comments and Suggestions for Authors
No further comment, though novelty of this study is still questionable.